# Formic Acid Dehydrogenation Using Noble-Metal Nanoheterogeneous Catalysts: Towards Sustainable Hydrogen-Based Energy

Abbas Al-Nayili [1], Hasan Sh. Majdi [2], Talib M. Albayati [3,*] and Noori M. Cata Saady [4]

1 Chemistry Department, College of Education, University of Al-Qadisiyah, Al Diwaniyah 58001, Iraq; abbas.al-nayili@qu.edu.iq
2 Department of Chemical Engineering and Petroleum Industries, Al-Mustaqbal University College, Babylon 51001, Iraq; hasanshker1@gmail.com
3 Department of Chemical Engineering, University of Technology Iraq, Baghdad 35010, Iraq
4 Department of Civil Engineering, Memorial University of Newfoundland, St. John's, NL A1B 3X5, Canada; nsaady@mun.ca
* Correspondence: talib.m.naieff@uotechnology.edu.iq

**Abstract:** The need for sustainable energy sources is now more urgent than ever, and hydrogen is significant in the future of energy. However, several obstacles remain in the way of widespread hydrogen use, most of which are related to transport and storage. Dilute formic acid (FA) is recognized asa a safe fuel for low-temperature fuel cells. This review examines FA as a potential hydrogen storage molecule that can be dehydrogenated to yield highly pure hydrogen ($H_2$) and carbon dioxide ($CO_2$) with very little carbon monoxide (CO) gas produced via nanoheterogeneous catalysts. It also present the use of Au and Pd as nanoheterogeneous catalysts for formic acid liquid phase decomposition, focusing on the influence of noble metals in monometallic, bimetallic, and trimetallic compositions on the catalytic dehydrogenation of FA under mild temperatures (20–50 °C). The review shows that FA production from $CO_2$ without a base by direct catalytic carbon dioxide hydrogenation is far more sustainable than existing techniques. Finally, using FA as an energy carrier to selectively release hydrogen for fuel cell power generation appears to be a potential technique.

**Keywords:** nanoheterogeneous catalysts; formic acid (FA); dehydrogenation; chemical hydrogen storage; hydrogen economy; sustainable energy



## 1. Introduction

It is widely acknowledged that non-renewable coal and oil may be used for a few decades. Transportation and other energy-related sectors will be highly needed as safe and sustainable energy carriers in the future. Researchers are working in various sectors such as geothermal power, lithium-ion batteries, solar energy conversion, and nuclear energy, all of which have the potential to solve the energy crisis in the foreseeable future [1–7]. Hydrogen ($H_2$) is a viable medium-term energy storage option. It is expected to contribute significantly to the future energy system as a supplementary fuel and energy carrier [8–10]. Hydrogen has a high specific energy of 33.30 kWh $kg^{-1}$ than diesel, which holds only about 12–14 kWh $kg^{-1}$. On the other hand, a hydrogen economy is unlikely to emerge unless major integrated technological advancements in $H_2$ generation, storage, and delivery systems are achieved [11]. Figure 1 summarizes the key advantages of hydrogen gas as an energy carrier [8,12].

Creating a safe and effective hydrogen storage system is a major problem [13,14]. Accordingly, various sophisticated research techniques to create novel materials that store and distribute hydrogen at satisfactory rates have been developed to utilize hydrogen as a green energy source and solve the challenges of its efficient and safe storage. Physical

or chemical storage can be classified according to the methods used. In the physical storage technique, $H_2$ can be held in its diatomic molecule state in a covered vessel under low temperatures and high pressure, such as in the case of cryo-compression and tanks of high pressure [15], or adsorbed on high surface area materials, such as metal-organic frameworks [3,16–19], clathrate hydrates [20], zeolites [21,22] and various carbon materials [23–26]. In the chemical storage technique, $H_2$ is stored in a chemically bonded state rather than in a molecular state. Typically, certain appropriate compounds have been selected because they have greater hydrogen content and can release hydrogen effectively at ambient conditions (temperature and pressure) by catalytic or non-catalytic methods. Examples of such compounds include hydrous hydrazine, metal amidoborates, metal borohydrides, ammonia borane, sodium borohydride and formic acid (HCOOH; FA) [27–31]. However, because of their poor kinetics for reversible hydrogen adsorption-desorption interactions, low intrinsic thermal conductivity, thermodynamic stability, toxicity, and high price, the practical applicability of several of these hydrogen storage compounds is greatly limited [32].

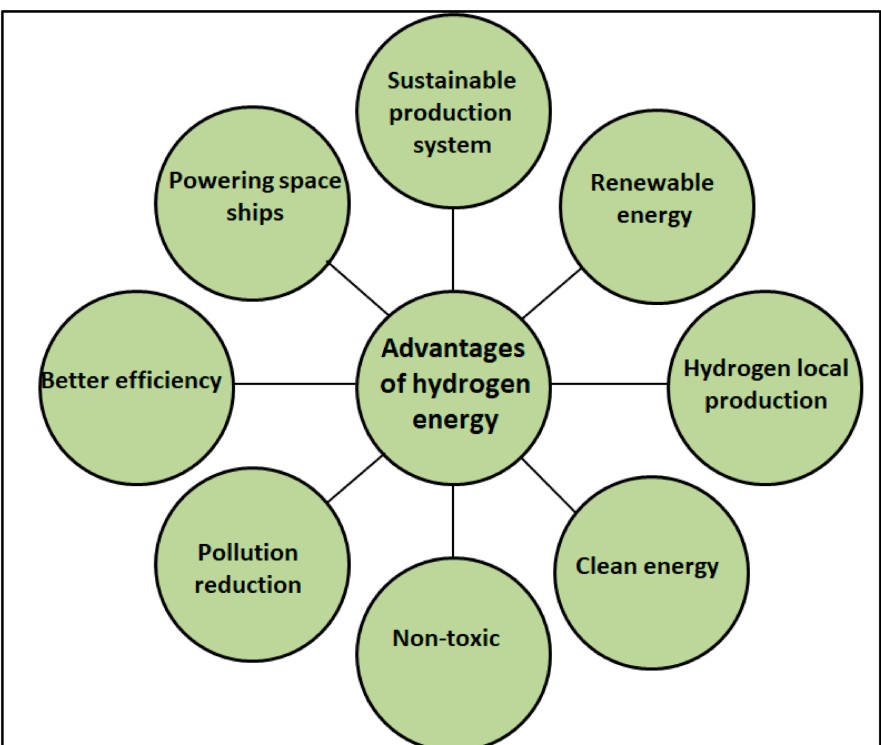

**Figure 1.** Advantages of hydrogen energy.

Organic liquid molecules such as FA have attracted interest because of their high energy density, low toxicity, and ease of handling [33,34]. Formic acid is one of the primary products generated during biomass processing, containing 4.4 wt.% hydrogen. In addition, pure FA contains 52 g $H_2$/L or 43.8 g $H_2$/kg, which makes it one of the important liquid organic hydrogen carriers (LOHC) [35]. Furthermore, FA transportation and refilling are simple because FA is liquid at room temperature; this allows it to be handled similarly to diesel and gasoline [36].

Formic acid is a highly promising hydrogen storage substance available nowadays. Although the hydrogen content of FA (4.40 wt.%) is lower than the US Department of Energy's goal in 2012 [37], it outperforms most other state-of-the-art hydrogen storage materials in terms of usable/net capacity. In addition to its advantages, FA is highly stable at ambient temperature without catalysts. Besides its inherent properties, another significant advantage of using FA as a hydrogen storage substance is that even the carbon dioxide ($CO_2$) produced during FA dehydrogenation can be hydrogenated later; this regenerates

FA molecules in a carbon-free release process. Moreover, FA can serve as a connection between renewable energy and hydrogen fuel cells. Progress can only be achieved with the right catalysts. This study concentrates on noble-metal heterogeneous catalysts for formic acid dehydrogenation at ambient temperature (20–50) °C (Figure 2).

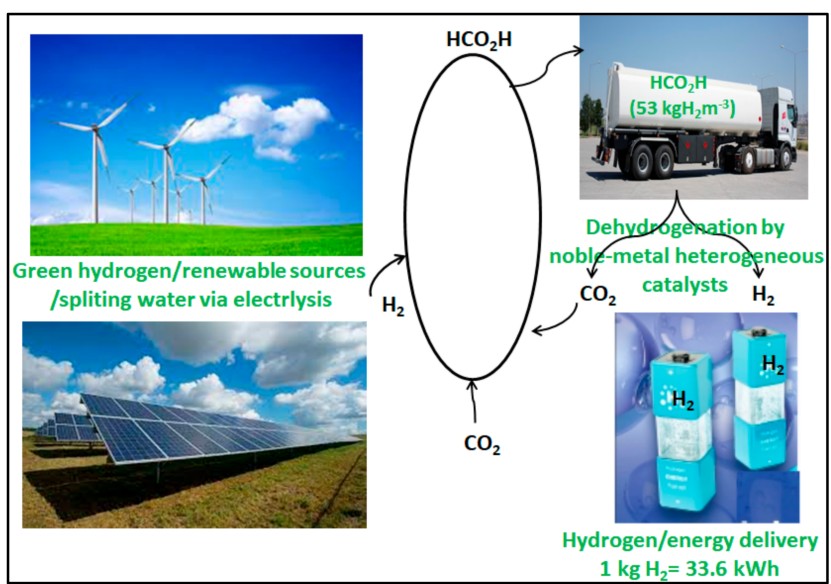

**Figure 2.** Carbon neutral energy storage via formic acid as a hydrogen carrier molecule.

Other liquid organic compounds, commonly known as liquid organic hydrogen carriers (LOHC) [38], such as methanol carbazole, cycloalkanes, and others, have been intensively explored in addition to formic acid. However, these compounds have several drawbacks that make them unsuitable as hydrogen storage materials, including toxicity, expense, poor stability, slow dehydrogenation kinetics, and low regeneration efficiency [39,40].

For the past ten years, the homogeneous or heterogeneous catalytic breakdown of FA to pure hydrogen has been a focus of research. The use of homogeneous catalysts to degrade FA at moderate temperature and pressure has been documented in several studies [41,42]. They reported encouraging findings in terms of catalyst stability and selectivity to $H_2$ and $CO_2$ and considerably increased catalytic efficiency. Isolation from the reaction mixture, moderate selectivity, the requirement for organic solvents or additives, and, in certain circumstances, severe reaction conditions hinder scaling up these catalysts for practical uses [34,43]. An alternative and appealing method is using heterogeneous catalysts, which can produce high catalytic activity (high substrate-to-metal molar ratio and high turnover frequency (TOF)) at low temperatures and with good selectivity towards $H_2$ [44,45]. As seen in reviews on various catalytic systems [4,32,46–51], the search for appropriate and efficient catalysts continues. Li et al. and Zhong et al. reviewed the dehydrogenation of aqueous formic acid by heterogeneous catalysts [32,52]. Doustkhah et al. focused on using Pd nanoalloys for hydrogen generation from formic acid [53]. This article reviews recent advances in using FA for chemical hydrogen storage, focusing on its dehydrogenation by metal nanoparticles as nanoheterogeneous catalysts with active noble metals such as Au and Pd at ambient temperature (20–50 °C) to obtain high catalytic activity and selectivity. Furthermore, we demonstrated that the low temperature required to create hydrogen from formic acid is a critical feature, as fuel cells are intended to provide energy to portable devices with low heat management profiles.

## 2. Formic Acid Decomposition

Fuel cells have remained unpopular because of the high cost of creating, storing, and transporting hydrogen. Rather than delivering hydrogen gas, having a chemical hydrogen

storage substance or hydrogen-containing substance which can be decomposed under ambient circumstances to create $H_2$ gas whenever needed is more practical.

Using a catalytic dehydrogenation process can release the hydrogen contained in FA through two main pathways (i) dehydrogenation/decarboxylation yielding $H_2$ and $CO_2$, and (ii) dehydration/decarboxylation yielding $H_2O$ and CO (Equations (1) and (2)) [54–56].

(i) dehydrogenation/decarboxylation

$$HCOOH \rightarrow H_2 + CO_2 \left( \Delta G^\circ_{298\ K} = -32.90\ kJ\ mol^{-1},\ \Delta H^\circ_{298\ K} = 31.20\ kJ\ mol^{-1},\ and\ \Delta S^\circ_{298\ K} = 216.0\ J\ mol^{-1}\ K^{-1} \right) \quad (1)$$

(ii) dehydration/decarbonylation

$$HCOOH \rightarrow H_2O + CO \left( \Delta G^\circ_{298\ K} = -12.40\ kJ\ mol^{-1},\ \Delta H^\circ_{298\ K} = 29.20\ kJ\ mol^{-1},\ and\ \Delta S^\circ_{298\ K} = 139.0\ J\ mol^{-1}\ K^{-1} \right) \quad (2)$$

The CO-free degradation of formic acid in the pathway (i) is essential for formic acid-based hydrogen storage, whereas pathway (ii) produces CO, which is an undesirable product that deactivates fuel cell catalysts. Therefore, pathway (ii) should be avoided. Based on this, and depending on the catalysts, reaction temperatures and pH values of the solutions, carbon monoxide (CO), a deadly poison to fuel cell catalysts, can also be produced by an undesired dehydration pathway (Equation (2)). High temperatures generally enhance dehydration during the reaction [57,58]. Fuel cell systems require ultrapure hydrogen to generate energy, and because CO poisons them, intake CO concentrations should be kept below 20 ppm to avoid long-term performance loss.

The dehydrogenation of FA (Equation (1)) produces only gaseous products ($H_2/CO_2$), with no accumulation or formation of by-products, making it advantageous compared to other alternative hydrogen carriers, particularly for portable usage. The gas combination produced is used as a feed-gas for an $H_2$/air fuel cell directly [59]. Because such fuel cells are intended to deliver energy to portable devices with limited heat management capabilities, mild temperatures are crucial.

Figure 3 presents a plausible mechanism for producing hydrogen from formic acid using metal nanoparticle (MN) catalysts. The following reactions would occur as a result of the mechanism:

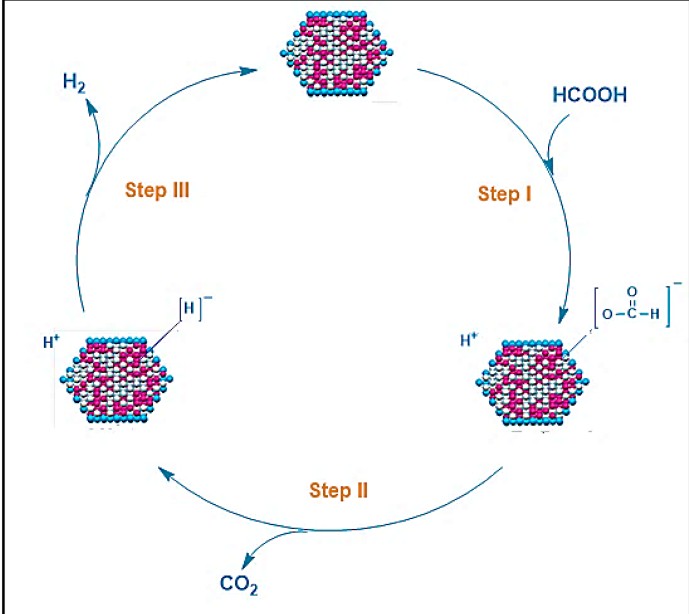

**Figure 3.** Reaction mechanism described for $H_2$ generation from FA dehydrogenation catalyzed by metal nanoparticle catalysts.

**Step I:** The cleavage of an O-H bond produces a proton ($H^+$) and a metal nanoparticle-formate (MN-[HCOO]$^-$) as an intermediate.

**Step II:** The C-H bond in the (MN-[HCOO]$^-$) species is broken to produce metal nanoparticle hydride (MN-[H]$^-$) and $CO_2$.

**Step III:** After the recombination of MN-[H]$^-$ with an $H^+$, $H_2$ is produced, and the MN species regenerate.

## 3. Heterogeneous Catalysts

Studying FA decomposition using heterogeneous catalysts has occurred since the 1930s, yet optimizing the catalysts, and measuring the CO produced by the FA dehydration-side reaction were not fully examined in early research [44]. In that state, the reaction was mostly examined in the gas phase, necessitating temperatures greater than 100 °C (formic acid's usual boiling point) or using an inert carrier gas that dilutes FA under its saturated vapor pressure; both would add to the setup's complexity. Because of their large surface area/mass ratio, nanoparticles are particularly well suited to act as catalysts in fuel cells. As a result, their utilization saves time and money. Developing heterogeneous catalysts for liquid-phase FA dehydrogenation is therefore crucially required [60]. A number of publications on nanoheterogeneous catalysts with different support materials for the dehydrogenation of aqueous FA have been published, emphasizing $H_2$ selectivity and low-temperature activity. Table 1 illustrates nanoheterogeneous catalysts used in the FA dehydrogenation reaction at ambient temperature (20–50 °C).

**Table 1.** Nanoheterogeneous catalysts to decompose aqueous formic acid at ambient temperature (20–50 °C).

| Catalyst | Temp. (°C) | Reagent | Turnover Frequency (TOF) ($h^{-1}$) | Ref. |
|---|---|---|---|---|
| Pd/C | 25 | Formic acid | 64 [b] | [61] |
| Pd/MSC-30 | 30 | Formic acid (0.6 M)/sodium formate (different molar ratios) | 1059 [a] | [62] |
| Pd/C | 25 | Formic acid (9.9 M)/sodium formate (3.3 M) | 304 [a] | [63] |
| Pd/mpg-$C_3N_4$ | 25 | Formic acid (1.0 M) | 144 [a] | [64] |
| PdIMP/CNF-HHT | 30 | Formic acid (0.5 M) | 563.2 [a] | [65] |
| PdSI/CNF-HHT | 30 | Formic acid (0.5 M) | 527.5 [a] | [65] |
| Ag@Pd/C | 20 | Formic acid (1.0 M) | 125 [c] | [66] |
| Au@Pd/N-mrGO | 25 | Formic acid (1.0 M) | 89 [a] | [67] |
| $Ag_{0.1}Pd_{0.9}$/rGO | 25 | Formic acid | 105 [a] | [35] |
| $Ag_{42}Pd_{58}$ | 50 | Formic acid (1.0 M) | 382 [a] | [68] |
| $Au_{41}Pd_{59}$/C | 50 | Formic acid (1.0 M) | 230 [a] | [69] |
| $Co_{0.30}Au_{0.35}Pd_{0.35}$ | 25 | Formic acid (0.5 M) | 80 [a] | [70] |
| CoAuPd/DNA–rGO | 25 | Formic acid (1.0 M) | 85 [a] | [61] |
| $Ni_{0.40}Au_{0.15}Pd_{0.45}$/C | 25 | Formic acid (0.5 M) | 12.4 [a] | [71] |
| Pd-Ni-Ag/C | 50 | Formic acid (0.175 M)/sodium formate (0.175 M) | 85 | [72] |
| $(Co_6)Ag_{0.1}Pd_{0.9}$/rGO | 50 | Formic acid (1 mol)/sodium formate (0.4 mol) | 2739 | [73] |

[a] Initial TOF values calculated for the initial stages of the catalytic reactions. [b] TOF calculated after 160 min. [c] TOF calculated after 60 min.

The majority of nanoheterogeneous catalysts decomposing aqueous FA to $H_2$ and $CO_2$, are supported by monometallic (single metal), bimetallic (two different metal elements), and trimetallic (three different metal elements) nanoparticles.

### 3.1. Monometallic Heterogeneous Catalyst

Several studies have been published recently on monometallic noble metal nanoparticles (NPs) based on different catalysts for FA dehydrogenation in an aqueous medium [62,74–76].

Bi et al. [77] used hyper-dispersed subnanometric gold NPs on $ZrO_2$ as catalysts to show the moderate and selective dehydrogenation of an FA/amine mixture. Under ambient conditions, the catalytic processes happen effectively and selectively (100%), with high TOFs/catalyst turnover numbers (TONs), and without generating any undesirable by-products such as CO. At ambient temperature, very efficient hydrogen was obtained from the production, from aqueous solution, of formic acid/sodium formate catalyzed via in situ-produced Pd/C with citric acid. Surprisingly, the addition of citric acid in the middle of the synthesis and development of Pd NPs on carbon improves the catalytically activity of the resultant Pd/C, on which the greatest conversion and turnover frequency for the breakdown of formic acid/sodium formate system can be achieved at ambient temperature [78]. Recently, well-dispersed Pd NPs (2.5 ± 0.3 nm) dropped on reduced graphene oxide sheets (rGO) via soil-immobilization methods have been prepared. Under moderate temperature (303 k), the resulting Pd/rGO-SI showed a high catalytic activity and selectivity for dehydrogenation of FA without undesired CO contamination. It is interesting to note that the initial value of TOF has reached 911 h$^{-1}$ (Figure 4) [79]. Zhu et al. created hyper-dispersed monometallic Pd NPs supported on MSC-30 (nanoporous carbon) which were synthesized via the sodium hydroxide-assisted reduction method. On MSC-30, the use of NaOH during particle production and development resulted in ultrafine Pd NPs that were well-dispersed. The resultant catalyst's performance was greatly improved by the combination of - contact and high NP dispersion. For heterogeneously catalyzed FA breakdown at 50 °C, this catalyst had very high selectivity of $H_2$ (100%) and activity (TOF = 2623 h$^{-1}$) [62].

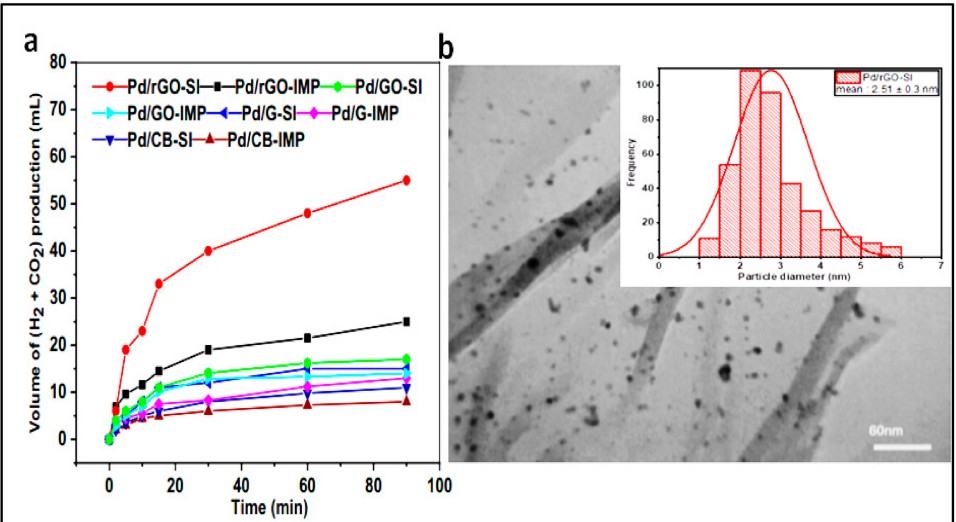

**Figure 4.** Formic acid decomposition over nanoheterogeneous Pd prepared by different methods, and TEM image: (**a**) Volume of generated gas ($H_2 + CO_2$) Reaction conditions: 0.12 mg of catalyst, 30 °C, 0.5 M FA, (substrate/metal molar ratio 2000:1), 800 rpm and 90 min reaction time. (**b**) TEM images and insert figure show distribution of particle size in Pd/rGO-SI. Reprinted with permission from ref. [79]. Copyright 2021 Springer Nature.

### 3.2. Bimetallic Heterogeneous Catalyst

It is generally known that adding a secondary metal to the active phase can change the electrical characteristics and adsorption behavior and the metal dispersion/particle size. Presently, supported Pd-based nanocatalysts are shown to be active for the dehydro-genation of aqueous FA [62–65,78]. Adding Au or Ag to Pd NPs in an aqueous medium significantly enhances their stability and catalytic activity [35,66–69]. The enhanced cat-alytic activity of bimetallic Pd-Au/C and Pd-Ag/C catalysts was attributable to the greater tolerance to CO poisoning of Ag and Au. The addition of $CeO_2(H_2O)x$ increased catalytic activity even more because $CeO_2$ forms cationic palladium species with strong activity in

CO oxidation [80] and methanol decomposition [81]. An alternative justification is that $CeO_2(H_2O)x$ on the Pd surface can trigger FA breakdown via a more effective mechanism, resulting in less poisoning intermediates [82]. Huang et al. developed a new Pd-Au bimetallic catalyst with a Pd-Au@Au core–shell nanostructure supported on carbon. They employed it to effectively catalyze hydrogen production by FA breakdown after being synthesized utilizing a simultaneous reduction process without the use of stabilizers. At low temperatures, the catalyst showed excellent activity, selectivity and stability, outperforming monometallic catalysts [83]. Tsang et al. created diverse core–shell nanoparticles with a metal element inner core and a palladium outer shell. At room temperature, Ag@Pd nanoparticles (diameter 8 nm) with 1–2 atomic layers of Pd shell had the maximum activity to break down FA, whereas comparable pure Pd and Ag/Pd alloy catalysts had extremely weak activity. At 20 °C, an equimolar $H_2$ and $CO_2$ mixture was constantly created with no evidence of CO. However, at temperatures greater than 50 °C, CO was observed [66,84]. Moreover, theoretical calculations revealed a firm link between the metal core's work function and the catalytic activity: the highest net difference with the Pd shell's work function led to excellent adsorption energy by charge transfer from its core to the shell. Therefore, the highest suitable activity of the resulting bimetallic structural system for FA decomposition was obtained. Yamashita et al. encapsulated Pd NPs via photo-assisted and ion exchange deposition techniques on metal-organic framework MIL-125, as well as its amine-functionalized equivalent $NH_2$-MIL-125 [85]. In relation to Pd-MIL-125 and different Ti-based porous materials, Pd-$NH_2$-MIL-125 exhibited impressive catalytic activity for $H_2$ production from FA at 30 °C (TOF = 214 $h^{-1}$). The fundamental Metal Organic Framework (MOF) functionalization and the small sizes of NP essentially dictated the remarkable catalytic performance. Furthermore, the photo-assisted deposition approach was acknowledged as being a remarkably successful method to create small and distributed NPs in MOF frameworks. The assembly of AgPd NPs on single-layer carbon material, e.g., reduced graphene oxide (rGO), was achieved utilizing a simple co-reduction approach, with the rGO acting as an influential dispersion agent and a unique sustenance for nanocatalysts [35]. With (105 $h^{-1}$) initial TOF, the $Ag_{0.1}Pd_{0.9}$/rGO produced achieved 100% $H_2$ selectivity and extremely strong activity toward full dehydrogenation of FA at ambient temperature [35]. Recently, the sol-immobilization approach has been used to effectively manufacture well-dispersed AuxPdy (Au/Pd: 3:1, 1:1, and 1:3) NPs supported on rGO. Under moderate circumstances, the Au-Pd/rGO produced a high catalytic activity and full selectivity for the dehydrogenation of FA without undesired CO contamination. The synergetic influence of Au–Pd nanostructures and the enhanced reaction sites equally dispersed over the rGO support are attributed to bimetallic catalyst's greater activity (Figure 5) [86].

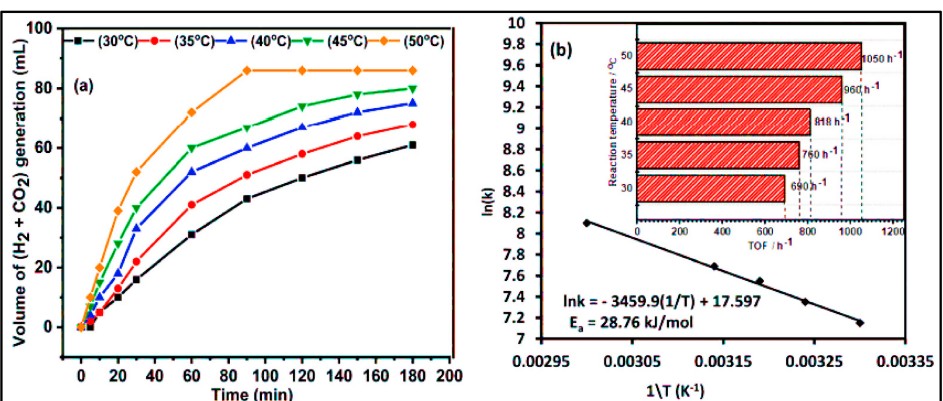

**Figure 5.** Volume of $H_2$ gas evolution from the aqueous FA solutin and the corresponding TOF: (**a**) Volume of ($H_2$ + $CO_2$) gas formed from formic acid decomposition vs. time, and (**b**) TOF of $H_2$ values produced with the $Au_1Pd_3$/rGO catalyst at (30–50 °C). Reproduced from ref. [87] with permission from the Royal Society of Chemistry.

Zhang et al. created well-distributed Pd-Ni nanocatalysts developed on a composite of GNs–CB (graphene nanosheet–carbon black) to incorporate the benefits of graphene nanosheets and carbon black [88]. Surprisingly, Pd-Ni NCs built on GNs–CB had better catalytic activity for FA breakdown in the aqueous phase at 25 °C than Pd or Ni did alone. Further efforts to improve and sustain the catalytic efficiency of catalysts in the advancement of FA as a hydrogen storage substance will benefit from using GNs–CB as a novel type of carbon support to distribute, promote and anchor nanocatalysts with more active components.

*3.3. Trimetallic Heterogeneous Catalyst*

Trimetallic NPs have lately acquired increased attention, particularly in catalytic systems, because of their novel physicochemical features (e.g., catalytic, electrical, optical, and magnetic) which are caused by their monometallic counterparts' synergistic effects [89]. Yurderi et al. used a simple and repeatable wet impregnation followed by a simultaneous reduction method at room temperature to synthesize Pd-Ni-Ag (trimetallic nanoparticles) with various metal ratios, as well as their Pd-Ni, Ni-Ag, and Pd-Ag (bimetallic) and Pd, Ni, and Ag (monometallic) counterparts, loaded on active carbon [72]. Under mild reaction conditions, all composites produced were used as nanoheterogeneous catalysts to break down FA. At 50 °C, Pd-NiAg/C catalyzed FA dehydrogenation with a 100% selectivity and a TOF of 85 h$^{-1}$ activity. Pd-Ni-Ag nanoparticles have outstanding ability to resist leaching, CO poisoning and agglomeration, enabling the reusability of Pd-Ni-Ag/C catalysts in FA dehydrogenation. Yan et al. also reported that a $Co_{0.30}Au_{0.35}Pd_{0.35}$ nanoalloy supported on carbon can generate hydrogen free of CO from FA dehydrogenation at 25 °C, and that it is a highly efficient, stable and low-costing catalyst [70]. Figure 6 shows the catalytic activity of $Co_{0.30}Au_{0.35}Pd_{0.35}/C$, its mono-metallic counterparts Pd/C, Au/C, and Co/C, and its bi-metallic counterparts $Au_{0.50}Pd_{0.50}/C$, $Co_{0.30}Pd_{0.70}/C$, and $Co_{0.30}Au_{0.70}/C$ for $H_2$ production from FA breakdown at 25 °C. The activity of the $Co_{0.30}Au_{0.35}Pd_{0.35}/C$ catalyst as prepared was significantly higher than that of the mono-metallic and bi-metallic catalysts prepared via the identical method [70].

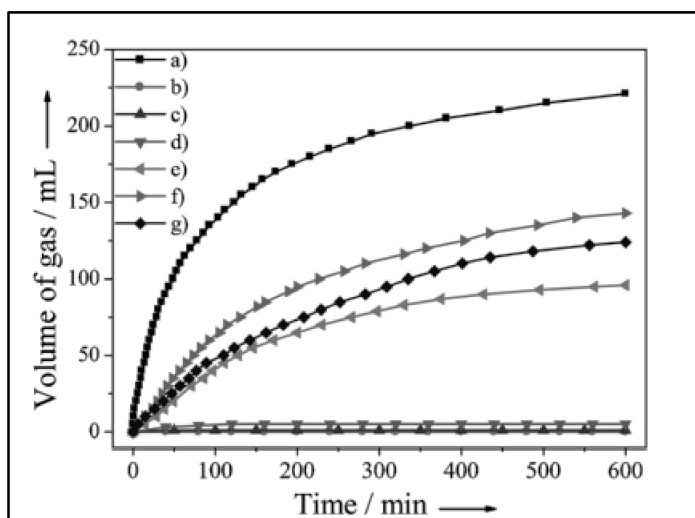

**Figure 6.** Gas generation by decomposition of FA (0.5 m, 10 mL) versus time in the presence of (a) Co 0.30 Au0.35Pd0.35 /C, (b) Co/C, (c) Au/C, (d) Co 0.30Au0.70 /C, (e) Pd/C, (f) Co 0.30 Pd0.70/C, and (g) Au0.50 Pd0.50 /C (nmetal /nFA = 0.02) at 298 K in ambient atmosphere. Reprinted with permission from ref. [70]. Copyright 2021 Wiley-VCH.

The catalytic trimetallic composite of Co-Au-Pd/DNA–rGO was synthesized using a DNA-directed and simple technique [61]. The FA dehydrogenation catalytic activity of Co-Au-Pd/DNA–rGO has been compared to that of Co-Au-Pd/rGO and Co-Au-Pd

NPs in the absence of additives at 25 °C. In comparison to other catalysts, the Co-Au-Pd/DNA–rGO composite has the greatest activity, and the rank of activity is Co-Au-Pd NPs < Co-Au-Pd/rGO < Co-Au-Pd/DNA–rGO. With DNA or DNA–GO composite, no CO gas was created from the FA aqueous solution, suggesting that DNA and DNA–GO function, respectively, as a template and support for the formation of Co-AuPd NPs rather than as catalytic agents for FA dehydration.

## 4. Formic Acid Fuel Cells (DFAFCs)

FA's hyper-gravimetric capability was recognized, and using FA as a secondary fuel in direct FA fuel cells (DFAFCs) was suggested and investigated [90]. While DFAFCs suffer from significant problems, hydrogen fuel cells perform a role in a well-established technology that has been commercialized in fuel cell vehicles (FCVs) with outputs of over 140 kW and ranges of over 600 km. As a result, producing $H_2$ selectively from FA to power hydrogen fuel cells is a potential strategy with a speedy time-to-market. As a traditional fuel, energy discharge includes FA consumption, resulting in a massive release of $CO_2$ (Figure 7).

Direct formic acid fuel cells work in the same way as other fuel cells. They create electricity by oxidizing FA and reducing $O_2$. FA and $O_2$ (or air) are supplied to the anode and cathode, respectively, in the electrochemical cell. Protons can pass across an electrolyte membrane [91].

The following are the DFAFC's direct anode, cathode, and total reactions [83]:

$$HCOOH \rightarrow CO_2 + 2H^+ + 2e^- \tag{3}$$

$$0.5O_2 + 2H^+ + 2e^- \rightarrow H_2O \tag{4}$$

$$HCOOH \rightarrow CO_2 + H_2O \tag{5}$$

The anode indirect reaction is:

$$HCOOH \rightarrow CO_{ads} + H_2O \rightarrow CO_2 + 2H^+ + 2e^- \tag{6}$$

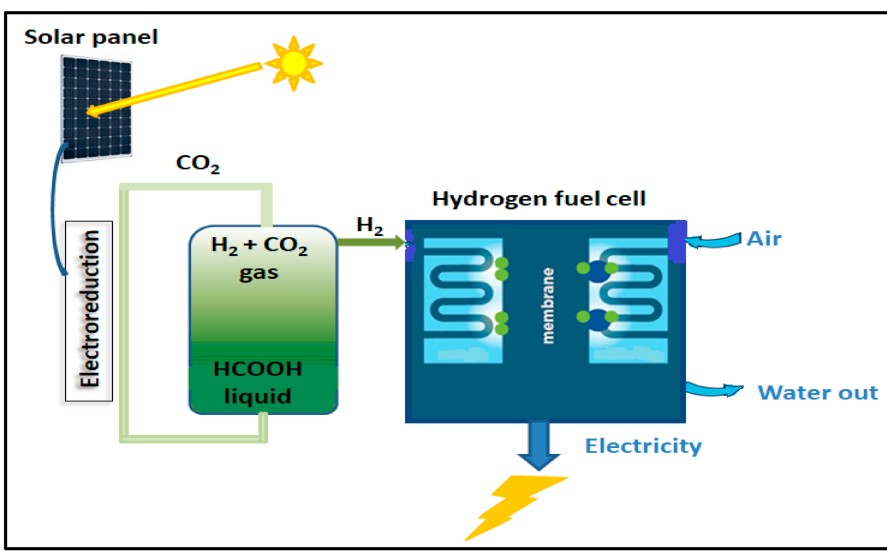

**Figure 7.** Basics of formic acid fuel cell and hydrogen energy.

Direct FA oxidation occurs at the anode through decomposition (Equation (3)) [85].

In the optimal direct pathway, $HCOOH \rightarrow *HCOO$ (formate) or $*COOH$ (carboxyl) converts fully into $H_2$ and $CO_2$ [86].

FA is dehydrated in an indirect manner (Equation (6)), $HCOOH \rightarrow *HCOO \rightarrow *CO \rightarrow CO_2$ by interacting with the active sites. However, the unwanted CO might poison the catalysts, increasing the overpotential needed for oxidation [85].

Theoretical [87,88] and experimental [89,90] studies have determined what occurred in the reaction.

A train of FA-based power can produce superior energy-to-mass ratios than existing fossil fuel-powered combustion engines when combined with a fuel cell and a lightweight electrical motor [92–94]. Furthermore, the cost of constructing and maintaining the distribution infrastructure is a significant barrier to large-scale consumer uses of $H_2$ gas [95].

Because FA is a harmless, ecologically friendly liquid with limited flammability at ambient temperatures, the current infrastructure of gasoline can be simply modified for FA delivery [96]. A catalytic FA converter system that creates onboard hydrogen to power a fuel cell in an automobile has unique criteria that must be met with respect to catalyst design [97,98]. Vital issues are (1) catalyst activity, as measured by the TOF (catalysts turnover frequency) at a given temperature, (2) $H_2$ selectivity, (3) catalyst lifetime or durability, as measured by the TONs (catalyst turnover numbers), and (4) the cost of the catalysts [99]. Economic factors heavily influence user acceptance of new technology. Any CO-generating activity must be controlled when FA is used as a chemical hydrogen carrier because CO formation affects total H2 yield and poisons the fuel cell's catalyst. CO poisoning of the Pt catalysts in a proton-exchange membrane fuel cell (PEMFC) is a key challenge in manufacturing industrial hydrogen fuel with typical vital CO levels >10 ppm [100–103]. As a result, any acceptable FA decomposition catalyst must have $10^5$ selectivity for dehydrogenation and dehydration. Heterogeneous transition metallic nanoparticle catalysts have very low selectivity and frequently create hydrogen with a high CO concentration > 1000 ppm. Enhanced hydrogen selectivity can be achieved with alloy nanoparticles [68,104–106].

Metal alloy nanoparticles, a single phased solid-solution mixture of two or more distinct metals, are a potential option for unpolluted Pt nanoparticles for hydrogen fuel cells. The nanoparticles of a metal alloy can give superior catalytic activity than monometallic nanoparticles because the synergistic effect of the catalytic activity of metal alloys is higher than the combined total of the individual metal components [107,108]. The strain and structure of the atoms at the metal alloy surface, which are important to the catalytic process, are modified when non-precious transition metals, such as Fe, Ni, or Co, are combined with Pt to generate Pt-based alloy nanoparticles [109,110]. This action alters catalytically active Pt's electrical and geometric structures and increases its catalytic activity [111,112]. The surface morphology of metal alloy nanoparticles depends strongly on their surface composition. It may be modified during alloy production by nanoparticle size and shape and the reaction temperature [108]. More crucially, while having a lower precious metal concentration, the metal alloy nanoparticles can maintain or increase the catalytic performance, allowing for commercial applications by addressing the underlying challenge of Pt in fuel cells (i.e., scarcity and high cost) [86,113].

## 5. Formic Acid Production

Formic acid is a crucial component created from a variety of chemical molecules. FA is found in the venom of ants in nature [92] and is emitted into the atmosphere as a result of forest emissions. Various chemical techniques can be used to prepare it. The most frequent industrial procedure is the synthesis of methyl formate from a mixture of carbon monoxide and methanol in the existence of a strong base at 80 °C and 40 atm, followed by hydrolysis of the methyl formate to yield FA [114]. FA can also be synthesized as a by-product of acetic acid production, biomass oxidation, $CO_2$ hydrogenation and biosynthesis via carbon dioxide reduction mediated by the enzyme formate dehydrogenase [115,116]. Despite the enormous progress that has been made in nano-chemistry and nanotechnology over the last two decades, and that forming FA from carbonates (primarily Pd-based) has been a subject of research for some time, few instances of supported metal catalysts at the nanometer

scale for direct hydrogenation of carbon dioxide have been reported. These primarily involve Au, Pd, and Ru. $CO_2$ hydrogenation through heterogeneous catalysts under mild reaction conditions will be the focus of this section. $CO_2$, a main result of chemical or electrochemical FA decomposition and the final product of organic substance combustion in air, is a significant greenhouse gas. Using $CO_2$ as a source for FA is not only a sensible way to create FA as a renewable energy source, but it also has the potential to benefit the environment. In 1935, the first report of $CO_2$ hydrogenation to FA was published, utilizing RANEY® nickel as a catalyst with hydrogen at 200–400 bar and 80–150 °C. To change the thermodynamic equilibrium synthesizing FA, amine has to be supplemented [117]. Sivanesan et al. used the Pd/C catalyst to reduce the carbonates to FA in moderate reaction conditions [118]. However, the chemical equilibrium between the formate and the carbonate prevented the reaction from proceeding to completion. Lee et al. [64] developed a reversible nanoheterogeneous (>3 nm) Pd catalyst based on Pd/mpg-$C_3N_4$ (mesoporous graphitic carbon nitride) for FA and $CO_2$ interconversion. This catalyst stimulates the generation of FA by $CO_2$ hydrogenation and FA dehydrogenation with a TOF of 144 $h^{-1}$, even in the absence of any external bases at 25 °C. In the presence of pristine $NEt_3$, Au black catalyzes $CO_2$ hydrogenation to create $HCOOH/NEt_3$ adducts. Using the high-boiling amine (n-$C_6H_{13})_3$N, FA is isolated from $HCOOH/NEt_3$ [119]. These discoveries, when combined with the catalytic FA breakdown to hydrogen free of CO and the readily removed and reused $CO_2$, perfect the chemical circle for the long-awaited $CO_2$-based hydrogen storage. When Pd NPs (ca. 4.4 nm) coated over g-$C_3N_4$ (graphitic carbon nitride) were equated with CNT-supported Pd NPs of comparable size, the catalytical activity on g-$C_3N_4$ was 12 times higher than on CNT. This investigation was conducted insde environmentally friendly solvent, namely water, in the absence of a base additive and in moderate circumstances (50 bar, 40 °C and $H_2/CO_2$ = 1). The great adsorption and activation of $CO_2$ capacity of g-$C_3N_4$, which delivers the active form of formate or carbonate to Pd when $H_2$ is activated, results in a considerable impact of the g-$C_3N_4$ support in contrast to CNT [120–122].

## 6. Conclusions and Perspectives

The benefits of a hydrogen economy are obvious, even if significant research is required to accomplish the essential technological advancements. Formic acid is an environmentally-benign hydrogen storage substance because of its easy storage and lack of poisonousness. Its production through dehydrogenation releases only gaseous products ($H_2/CO_2$). Interestingly, $CO_2$ can be converted back to formic acid using catalysts under moderate conditions, resulting in a $CO_2$-neutral hydrogen storage cycle. Noble metals, such as Au and Pd, can serve as nanoheterogeneous catalysts that work in aqueous formic acid solutions and ambient temperature (20–50 °C), were reviewed in this review. The Pd nanoparticles are employed in most nanoheterogeneous catalysts used in the formic acid dehydrogenation process. However, chemical intermediates adsorb on the nanoparticle surfaces and deactivate Pd monometallic systems. The situation with heterogeneous formic acid decomposition catalysts is identical to that with homogeneous systems. While the activity and $H_2$ selectivity have not yet been achieved homogeneous system levels and most heterogeneous systems tested still have some degree of decarbonylation activity, this gap is narrowing. The recent use of state-of-the-art nanoparticle synthesis techniques has resulted in a variety of high-performance catalysts, including bimetallic and trimetallic Pd and Au combinations that produce high-quality $H_2$ with minimal CO concentration. The direct formic acid fuel cell (DFAFC) example in this article marks significant progress toward prototype development, scale-up, and commercialization. Furthermore, $CO_2$ may be converted back to formic acid using catalysts under moderate conditions, resulting in a $CO_2$-neutral hydrogen storage cycle. More research is needed to make further advances, particularly for mobile applications.

**Author Contributions:** Conceptualization, A.A.-N. and T.M.A.; methodology, A.A.-N.; software, H.S.M.; validation, A.A.-N., H.S.M. and H.S.M.; formal analysis, H.S.M.; investigation, A.A.; resources, A.A.-N.; data curation, N.M.C.S.; writing—original draft preparation, N.M.C.S.; writing—review and editing, A.A.-N. and N.M.C.S.; visualization, T.M.A.; supervision, A.A.-N.; project administration, A.A.-N.; funding acquisition, H.S.M. All authors have read and agreed to the published version of the manuscript.

**Funding:** This research received no external funding.

**Data Availability Statement:** All relevant data are included in the paper.

**Acknowledgments:** The authors are grateful to the Department of Chemical Engineering at the University of Technology-Iraq, the Chemistry Department, College of Education, University of Al-Qadisiyah, Al Diwaniyah, Iraq, the Department of Chemical and Petroleum Industries Engineering, Al-Mustaqbal University College, Babylon, Iraq, and the Department of Civil Engineering, Memorial University of Newfoundland, St. John's, NL A1B 3X5, Canada.

**Conflicts of Interest:** We certify that the authors have no affiliations with or involvement in any organization or entity with any financial interest or nonfinancial interest in the subject matter or materials discussed in this manuscript.

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
