# Peer review of "Formic Acid Dehydrogenation Using Noble-Metal Nanoheterogeneous Catalysts: Towards Sustainable Hydrogen-Based Energy"

_catalysts, doi:10.3390/catal12030324_

Round 1

Reviewer 1 Report

The topic is of great interest and in my opinion this work could deserve to be published after major revision. I have found some aspects that need improvement. The most important ones are listed below:
1. In general, it gives the impression that the article has been written in a hurry and neglecting the presentation. For example, Figure 2 and equations i and ii are not publishable and need to be improved.
2. The references are very old, it is necessary to include, at least in the introduction, current references.
3. The concept monometallic, bimetallic and trimetallic catalysts is used erroneously throughout the manuscript. To avoid confusion, it should be clarified that these are heterogeneous catalysts that in the active phase are composed of a single type of metal or of two or three different types of metals. But they are not monometallic, bimetallic or trimetallic catalysts, that is completely false.

Reviewer 2 Report

Review report:

In this review, the authors reviewed the dehydrogenation of formic acid involving nanoheterogeneous catalysis. Several previous research results have been summarized and compared. This research field has attracted increasing research interests and developed rapidly. However, unfortunately, I could not recommend publishing it in its current form due to the following issues.

  1. There are many similar reviews that have been published in the past few years (Int. J. Hydrogen Energy, 2018, 43, 7055; Adv. Sust. Sys., 2018, 2, 1700161; Adv. Mat., 2020, 32, 2001818; Adv. Energy, Mat., 2019, 9, 1801275; Front. Mater., 2019, 6, 44; ACS Appl. Nano Mater., 2020, 3, 22; Chem. Pap., 2018, 72, 2121). However, the authors did not clarify the difference and novelty of their review from those previously published ones.

  1. More importantly, the references in this review are mainly in the years from 2009 to 2015 which is out of date. This research field has developed rapidly, the authors should review the state-of-the-art research results (at least within the past five years) so that the readers can know what is happening in this research field. 

  1. Some of the data in the review is inaccurate, for example in Table 1, the TOF of Ref. 43 should be 1059 h-1 at 30ºC rather than 2623 h-1. The author should carefully check their data in this review. Moreover, the authors are encouraged to include more information in this Table such as the concentration of formic acid, the additives, whether the TOF is an overall TOF or initial TOF, whether the TOF is calculated based on all the metal catalysts or only based on the surface active sites.

Reviewer 3 Report

The paper is not a full review but a mini-review on the subject. This manuscript required major revisions before being accepted for publication. Point to point comments are as follows:

1- Abstract can be more precise and focus on the objective and outcomes of the review.

2- In the introduction, the authors need to clearly state the novelty, contribution of the review. 

3. Clearly stated the scope of the review? why this review is required.

4-. Improve the quality of Fig 1 with better arrangement 

5. Fig 2 References, sources should be provided

6. Formic Acid decomposition reaction mechanism should be provided based on catalyst

7. Various types of catalyst tables should be provided expanded the literature 

8. Formic acid fuel cell section must be expanded and add mechanisms and working principle 

9. Grand table must be added in Section 4 and updated with modern techniques for FA production 

10. Conclusions must be based on object8ves and recommendations should be realistic. 

Reviewer 4 Report

This review paper reports on the “Dehydrogenation of Formic Acid Implementing Nanoheterogeneous Catalysis: Towards a Hydrogen-Based Economy as a Sustainable Energy”

To overall, I do think that this review is very interesting but there should be further modification before accepting this work for the publication in catalysts.

  1. There are many other materials (using sustainable energy) which could be used as source of hydrogen, authors must discuss on the advantages of formic acid compared to other compounds? The other important issue is whether the released hydrogen could be easily stored?

  1. The photos in some figures must be checked for the copy right, as they are, I suppose, taken from internet without declaring it.

  1. the entire text must be modified again to remove repeated sentences and text and give a clears picture on “what do this review paper add to our knowledge?

Round 2

Reviewer 1 Report

In my opinion, the revised version of the manuscript can be accepted for publication in catalysts

Author Response

Manuscript ID: catalysts-1584506

Type of manuscript: Review

Title: Formic Acid Dehydrogenation by Noble-Melal Nanoheterogeneous Catalysts: Towards Sustainable Hydrogen-Based Energy

1) Please add more recent references about the topic of the Review Article.

Thank you very much for your kind and encouraging comment.

1) Ans: These references are added.

  1. Nguyen, K.H. and M. Kakinaka, Renewable energy consumption, carbon emissions, and development stages: Some evidence from panel cointegration analysis. Renewable Energy, 2019. 132: p. 1049-1057.
  2. Onishi, N., et al., Development of effective catalysts for hydrogen storage technology using formic acid. Advanced Energy Materials, 2019. 9(23): p. 1801275.

  1. Li, X., et al., Metal–organic frameworks as a platform for clean energy applications. EnergyChem, 2020. 2(2): p. 100027.

  1. Lang, C., Y. Jia, and X. Yao, Recent advances in liquid-phase chemical hydrogen storage. Energy Storage Materials, 2020. 26: p. 290-312.

  1. Stucchi, M., et al., Synergistic Effect in Au-Cu Bimetallic Catalysts for the Valorization of Lignin-Derived Compounds. Catalysts, 2020. 10(3): p. 332.

  1. De Blasio, N., et al., Mission Hydrogen. 2021.

  1. van Renssen, S., The hydrogen solution? Nature Climate Change, 2020. 10(9): p. 799-801.

  1. Hafeez, S., et al., Experimental and Process Modelling Investigation of the Hydrogen Generation from Formic Acid Decomposition Using a Pd/Zn Catalyst. Applied Sciences, 2021. 11(18): p. 8462.

  1. Nie, W., et al., An amine-functionalized mesoporous silica-supported PdIr catalyst: boosting room-temperature hydrogen generation from formic acid. Inorganic Chemistry Frontiers, 2020. 7(3): p. 709-717.

  1. Wen, C., et al., A first study of the potential of integrating an ejector in hydrogen fuelling stations for fuelling high pressure hydrogen vehicles. Applied energy, 2020. 260: p. 113958.

  1. Xiao, R., et al., Effects of cooling-recovery venting on the performance of cryo-compressed hydrogen storage for automotive applications. Applied Energy, 2020. 269: p. 115143.

  1. Giappa, R.M., et al., A combination of multi-scale calculations with machine learning for investigating hydrogen storage in metal organic frameworks. International Journal of Hydrogen Energy, 2021. 46(54): p. 27612-27621.
  2. Barnett, B.R., et al., Observation of an Intermediate to H2 Binding in a Metal–organic Framework. Journal of the American Chemical Society, 2021. 143(36): p. 14884-14894.

  1. Gupta, A., et al., Hydrogen clathrates: Next generation hydrogen storage materials. Energy Storage Materials, 2021. 41: p. 69-107.

  1. Farajzadeh, M., et al., Anchoring Pd-nanoparticles on dithiocarbamate-functionalized SBA-15 for hydrogen generation from formic acid. Scientific Reports, 2020. 10(1): p. 1-9

  1. Nechaev, Y.S., et al., On the real possibility of “super” hydrogen intercalation into graphite nanofibers. Fullerenes, Nanotubes and Carbon Nanostructures, 2022: p. 1-9.

  1. Wang, C., et al., New strategies for novel MOF-derived carbon materials based on nanoarchitectures. Chem, 2020. 6(1): p. 19-40.

  1. Zheng, J., et al., Current research trends and perspectives on solid-state nanomaterials in hydrogen storage. Research, 2021. 2021.

  1. Lee, S.-Y., et al., Recent Progress Using Solid-State Materials for Hydrogen Storage: A Short Review. Processes, 2022. 10(2): p. 304.

  1. Zheng, J., et al., Current research trends and perspectives on solid-state nanomaterials in hydrogen storage. Research, 2021. 2021.

  1. Lee, S.-Y., et al., Recent Progress Using Solid-State Materials for Hydrogen Storage: A Short Review. Processes, 2022. 10(2): p. 304.

  1. Hoelzen, J., et al., Hydrogen-powered aviation and its reliance on green hydrogen infrastructure–Review and research gaps. International Journal of Hydrogen Energy, 2021.

  1. Mardini, N. and Y. Bicer, Direct synthesis of formic acid as hydrogen carrier from CO2 for cleaner power generation through direct formic acid fuel cell. International Journal of Hydrogen Energy, 2021. 46(24): p. 13050-13060.

  1. Liu, J., et al., Facile synthesis of agglomerated Ag–Pd bimetallic dendrites with performance for hydrogen generation from formic acid. International Journal of Hydrogen Energy, 2021. 46(9): p. 6395-6403.

  1. Park, J.H. and H.S. Ahn, Electrochemical synthesis of multimetallic nanoparticles and their application in alkaline oxygen reduction catalysis. Applied Surface Science, 2020. 504: p. 144517.

  1. Yang, S., et al., Enhancements in catalytic activity and duration of PdFe bimetallic catalysts and their use in direct formic acid fuel cells. Journal of Industrial and Engineering Chemistry, 2020. 90: p. 351-357.

  1. Sui, L., et al., Bimetallic Pd-Based surface alloys promote electrochemical oxidation of formic acid: Mechanism, kinetics and descriptor. Journal of Power Sources, 2020. 451: p. 227830.

  1. Valdes-Lopez, V.F., et al., Carbon monoxide poisoning and mitigation strategies for polymer electrolyte membrane fuel cells–A review. Progress in Energy and Combustion Science, 2020. 79: p. 100842.

  1. Ma, Z., et al., From CO2 to formic acid fuel cells. Industrial & Engineering Chemistry Research, 2020. 60(2): p. 803-815.

  1. Sivanesan, D., et al., Facile hydrogenation of bicarbonate to formate in aqueous medium by highly stable nickel-azatrane complex. Journal of Catalysis, 2020. 382: p. 121-128.

2) The title of the manuscript should be changed considering that the authors
focused mainly on noble metals base catalysts.

Ans: Thank you for your comment.

Formic Acid Dehydrogenation by Noble-Melal Nanoheterogeneous Catalysts: Towards Sustainable Hydrogen-Based Energy

Reviewer 2 Report

The references are still out of date in the revised manuscript which is lack of novelty compared with those previously published reviews.

Author Response

(The authors gave the same response as above.)

Reviewer 3 Report

The authors didn't expand the literature in two sections as I have indicated. I am clearly aware of the scope of review but a lots of references have been missed. Two comments were ignored by authors. For me both comments are as important as the other ones.  List of the Mono, bi and tri metallic catalyst for the said application is missing. 

Author Response

(The authors gave the same response as above.)
